neuroscience, cognition

numerosity perception, adaptation, confidence, reaction times, decision-making

**Author for correspondence:**
David C. Burr
e-mail: davidcharles.burr@unifi.it

# Adaptation to hand-tapping affects sensory processing of numerosity directly: evidence from reaction times and confidence

Paula A. Maldonado Moscoso[1], Guido M. Cicchini[2], Roberto Arrighi[1] and David C. Burr[1,2]

[1]Department of Neuroscience, Psychology, Pharmacology and Child Health, University of Florence, Florence, Italy
[2]Institute of Neuroscience, National Research Council, Pisa, Italy

PAMM, 0000-0001-9599-1246; GMC, 0000-0002-3303-0420; RA, 0000-0002-5435-6729; DCB, 0000-0003-1541-8832

Like most perceptual attributes, the perception of numerosity is susceptible to adaptation, both to prolonged viewing of spatial arrays and to repeated motor actions such as hand-tapping. However, the possibility has been raised that adaptation may reflect response biases rather than modification of sensory processing. To disentangle these two possibilities, we studied visual and motor adaptation of numerosity perception while measuring confidence and reaction times. Both sensory and motor adaptation robustly distorted numerosity estimates, and these shifts in perceived numerosity were accompanied by similar shifts in confidence and reaction-time distributions. After adaptation, maximum uncertainty and slowest response-times occurred at the point of subjective (rather than physical) equality of the matching task, suggesting that adaptation acts directly on the sensory representation of numerosity, before the decisional processes. On the other hand, making reward response-contingent, which also caused robust shifts in the psychometric function, caused no significant shifts in confidence or reaction-time distributions. These results reinforce evidence for shared mechanisms that encode the quantity of both internally and externally generated events, and advance a useful general technique to test whether contextual effects like adaptation and serial dependence really affect sensory processing.

## 1. Introduction

Perceptual adaptation is a form of short-term plasticity, usually generated by observing for some time a particular stimulus, such as a steadily drifting pattern. Adaptation has proved to be a fundamental psychophysical tool to study many perceptual properties, including high-level properties such as face identity and expression [1–3]. It has also proved invaluable in the study of the perception of *numerosity*, bringing this field of cognitive research into the realm of perceptual research [4–6]. Recently, cross-modal and cross-format adaptation have been used to demonstrate a 'generalized sense of number', showing strong interactions between the numerosities of spatial arrays of objects and temporal sequences of events [7]. Even more intriguingly, the authors went on to show interactions between numerosity perception and motor action: fast tapping reduces the apparent numerosity of both temporal sequences and spatial arrays, while slow tapping has the opposite effect [8].

These results are clearly important as they point to specific neural interactions between different forms of numerosity representation, reinforcing the neurophysiological evidence reported in macaque monkeys [9]. They also show strong neural links between numerosity and motor action, again with

parallels in the neurophysiological literature [10]. But do adaptation studies truly reveal underlying neural mechanisms as Mollon [1, p. 479] claimed (if you can adapt it it's there)? Can we think of adaptation as the 'psychologists microelectrode', as suggested by Frisby [11]?

It has recently been questioned whether adaptation necessarily reveals underlying neural mechanisms, with suggestions that they could result from changes in observer criteria, driven by cognitive, decisional processes, particularly for certain 'high-level' after-effects (for discussion see [12]). To demonstrate this possibility, Morgan *et al.* [13] showed that observers could simulate the effects of adaptation by adopting simple decision rules, along the lines of 'if unsure say fewer'. This strategy resulted in a clear shift of psychometric functions, without broadening the width of the functions (reflecting preserved precision). Therefore, it is possible that in the numerosity adaptation experiments the changes in the psychometric functions do not reflect changes in neural representations of number, but in a cognitive, decision strategy in reporting numerosity. Possibly after rapid tapping, there is a tendency to report uncertain numerosities as low, and after slow tapping to report these as high. This could conceivably account for the changes in apparent numerosity, without invoking the action on neural mechanisms.

Morgan *et al.*'s idea can be illustrated with a simple simulation shown in figure 1. The red curve illustrates a typical psychometric function, modelled by a cumulative Gaussian error function. The blue curve illustrates a hypothetical function of subjective confidence, based on the consistency of participant responses: one when certain, zero when guessing. On the basis of data from this study (figure 3), we assume minimal confidence is 50%, but this is not essential to the demonstration. Confidence should be minimal at the point of subjective equality (PSE), where sensory information is least. The green curve is the simulation of the strategy 'if unsure say fewer' (the product of the two probability functions), causing a downward shift of the curve, which necessarily shifts the function rightwards. The downward shift in the curve is virtually indistinguishable from a rightward shift caused by sensory adaptation to numerosity. However, if it is confidence that drives the downward shift, the confidence function itself should not change, but remain centred at the PSE of the unadapted function.

Gallagher *et al.* [14] took advantage of this fact to propose a novel way of distinguishing between sensory effects in adaptation and higher-level decisional biases, based on the assumption that confidence in the perceptual decision will scale with the strength of sensory evidence. In the typical two-alternative matching experiment used to measure adaptation, where participants choose which of two stimuli was the largest, the strength of sensory evidence will be weakest when their internal representations of magnitude are the same: that is, at the PSE. Therefore, the PSE should also correspond to the point of minimal confidence. If the PSE shifts with adaptation-induced changes in internal representations of magnitude, the shift in PSE should be accompanied by a comparable shift in minimal confidence. If, on the other hand, the adaptation results from weak confidence and a decision rule (as simulated in figure 1), the confidence ratings should remain minimal at the point of physical equality, and not shift with adaptation. Gallagher *et al.* [14] showed that adaptation to visual motion shifted not only the point of perceived equality of motion, but also the point of maximal

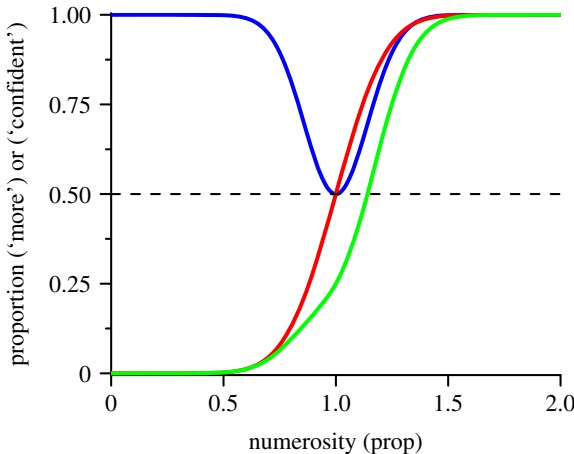

**Figure 1.** Simulation showing how response biases could induce a shift in psychometric function resembling a real sensory change. The red curve shows a hypothetical psychometric function for a numerosity discrimination task. The blue curve plots confidence level based on the relative numerosity difference between the stimuli. The green curve shows the result of a decision strategy 'less if unconfident', obtained by the pointwise product of two functions. (Online version in colour.)

decisional uncertainty. On the other hand, instructing participants to introduce a systematic response bias (along the lines of replicating Morgan *et al.*'s experiment) did not shift the point of maximal uncertainty.

Another common tool in sensory research is reaction times, which also vary systematically with sensory strength, well approximated by a power function of the stimulus strength plus a constant (Piéron's Law: [15]). Following the same logic discussed above, reaction times should also vary on a two alternative forced-choice task, being maximal when the sensory representations of the two are most similar, at the point of subjective equality. Therefore, adaptation should also shift the peak in reaction times, following the shift in PSE, if the effects are sensorial rather than decisional. If they remain anchored at physical equality, the adaptation is more likely to reflect response or decision biases.

In this study, we investigate how adaptation to numerosity affects confidence ratings and reaction times. We study two types of adaptation: visual adaptation to dense dot arrays [4], and motor adaptation to fast and slow hand-tapping [8]. The results show that both types of adaptation cause concomitant changes in both minimal confidence and maximal reaction times, suggesting that the effects of both adaptation to high-numerosity and to manual tapping are sensory rather than biases in decision.

## 2. Methods

Stimuli were presented on an Acer LCD monitor (screen resolution of 1920 × 1080, refresh rate 60 Hz) subtending 50° × 29° at the subject view distance of 57 cm. They were created with PsychToolbox routines for Matlab (v. R2016a, the Mathworks, Inc.) on a PC computer running Windows 7. In the motor adaptation conditions, hand movements were monitored by an infrared motion sensor device (leap motion controller—https://www.leapmotion.com) running at 60 Hz.

We used a standard forced-choice paradigm (figure 2). Stimuli were brief (250 ms) patches of dots, presented sequentially to the left and right of fixation, with a 200 ms pause between them. Each patch covered a circular region of 8° in diameter, centred at

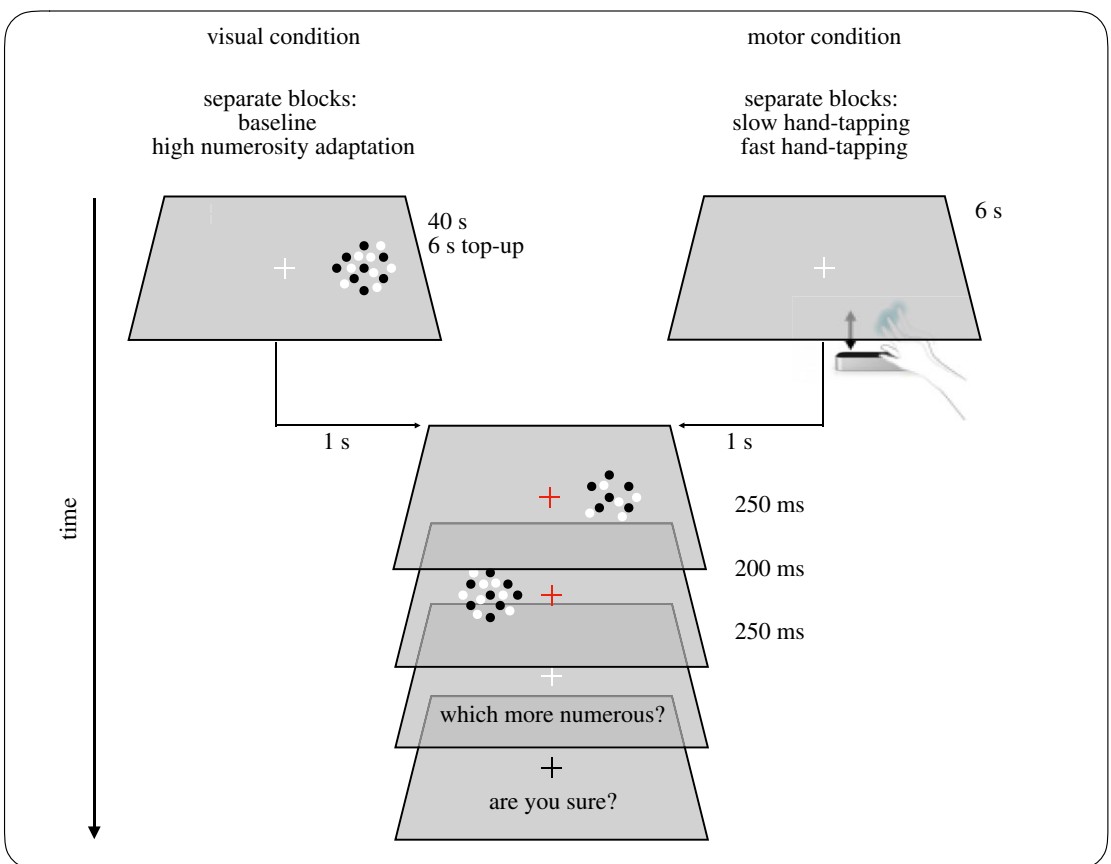

**Figure 2.** Stimuli and procedure. On each trial, subjects were required to indicate which of two stimuli was more numerous, then report whether they were confident with their response (both responses two alternative forced choice). In the visual adaptation condition, a dense dot array was displayed first for 40 s for 6 s top-up periods at the test location before the discrimination task (top left). In the motor adaptation condition (top right), participants were required to tap their hand with index finger extended, for 6 s on the right side of the screen, with their hand concealed by the screen and without touching any surface to minimize sensory feedback. Subjects either tapped as fast as possible or slowly, at around 1 Hz (tested in separated sessions). In all conditions, reaction times between the offset of the reference and the numerosity response were measured, although participants were never requested to make any speeded response. (Online version in colour.)

7° from screen centre. Dots were 0.3° diameter, separated from each other by at least 0.25°, half white and half black (to balance luminance), presented on a grey background. The patch to the left of fixation was the reference, with numerosity fixed at 16 dots; that to the right was the probe, with numerosity varying randomly from 8 to 32 dots (numerosity drawn from linear rectangle distribution). Participants first judged whether the stimulus on the left or the right appeared more numerous, then indicated their confidence in the judgements by pressing the up or down arrow (low or high confidence, respectively). We also measured the reaction times of the numerosity judgements, and report the mean, after removing outliers (more ±3 s.d. from the mean).

### (a) Adaptation

For the visual adaptation experiment, 12 participants (11 naive to the purpose of the study and one author; mean age 28 with normal or corrected-to-normal vision) adapted to an array of 60 dots (adapt to high) at the same position as the probe stimulus, for 40 s at the beginning of each session, then for 6 s top-up periods. Stimuli were presented 1 s after adaptation. Each participant performed a total of 432 trials. For the adaptation-to-tapping experiment, participants (nine naive to the purpose of the study and one author; mean age 28 with normal or corrected-to-normal vision) made a series of hand-tapping movements (pivoting at the wrist) on the right side of the screen until a white central fixation point turned red (the stop signal); 1 s later the stimuli were presented. In one condition participants tapped as rapidly as possible, in another at around 1 Hz. The program continuously monitored tapping via the infrared motion sensor: if a tap occurred after the presentation

of the test stimulus, the trial would be aborted. After the stimuli presentation, subjects were required to press the left arrow when the stimulus at left was perceived as more numerous, or the right arrow when the right-hand stimulus was perceived as more numerous. They then pressed the up-arrow if they were confident about the numerosity response or the down-arrow if they were not. Participants were unaware that we also measured the reaction time of the numerosity response, and they were not explicitly asked to make speeded responses. Three blocks of 24 trials were run for each condition.

### (b) Manipulation of rewards

We devised a control experiment to compare with adaptation, where we manipulated the reward rules. Ten adults participated in this study, nine naive to the purpose of the study (mean age 28 with normal or corrected-to-normal vision). Here, there was no adaptation, but participants played a point-based game, with three types of reward regimes (in different blocks). In baseline blocks, they received 1 point for each correct response and lost 1 for every error (performing on average at 85% correct). In 'reward-low' blocks, they received 2 points for correctly responded 'less than', and lost 1 for each error; and in 'reward-high', 2 points for correctly responding 'greater than', losing 1 for an error. They also indicated by pressing the up-arrow if they were confident about the numerosity indicated was 'less' or 'greater than' or the down arrow if they were not. They were given feedback on earning 50 points, and again at 80 points. Three blocks with at least 79 trials were run for each condition. We also measured the reaction time of the response, and again participants were not explicitly asked to make speeded responses.

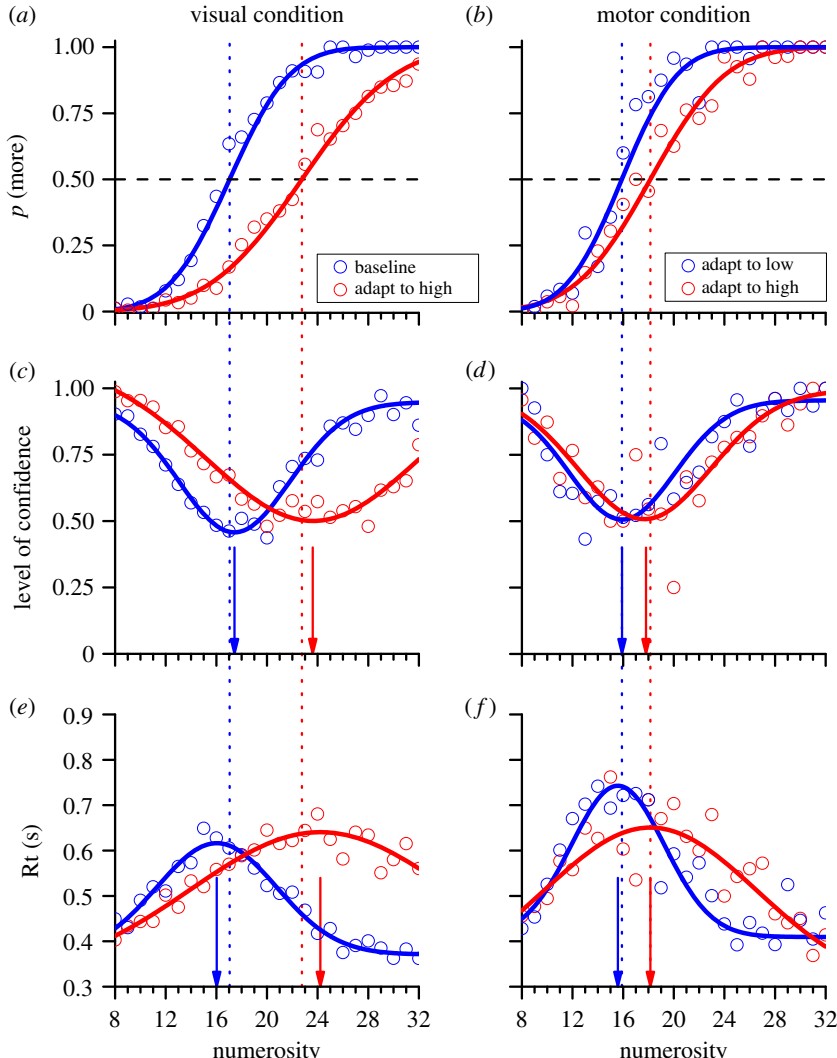

**Figure 3.** (a,b) Psychophysical functions showing proportion of trials in which the test was perceived more numerous than the reference, as a function of test numerosity. (c,d) Confidence levels and mean reaction times (Rt; (e,f)) as a function of test numerosity, for visual and motor adaptation (left and right panels, respectively). In all graphs, blue and red curves indicate baseline and high adaptation for visual adaptation (panels on the left-hand side) and slow or fast tapping in the motor experiment (on the right-hand side). The dashed lines show the PSEs and arrows the peaks of the best-fit Gaussians to the confidence or reaction-time distributions. (Online version in colour.)

## (c) Data analysis

The proportion of trials where the test appeared more numerous than the probe was plotted against physical numerosity and fitted with cumulative Gaussian error functions. The median of the error functions estimates the PSE, and the difference in numerosity between the 50% and the 75% points gives the just notable difference (JND). The distributions of average confidence responses (1 for high, 0 for low) and of the mean of reaction times were fitted with Gaussian distributions, and the peak of the fitted functions was taken as the point of maximum uncertainty or reaction times:

$$P(N) = b + a \cdot \exp\left(\frac{-(\bar{N} - N)^2}{2\sigma^2}\right), \tag{2.1}$$

where $N$ is numerosity, $P(N)$ the proportion of confident responses—or the average reaction time—at that numerosity, $b$ and $a$ constants, $\bar{N}$ the mean of the Gaussian and $\sigma$ the standard deviation. When fitting data pooled over participants, all parameters were free to vary. When fitting individual participant data, $b$ and $\sigma$ were fixed to the values obtained for the aggregate data.

All analyses were performed both on the 'aggregate participant', pooling all data from all participants, and also on individual

participant data. Significance of the aggregate data was calculated by bootstrap sign test: 10 000 reiterations, with replacement.

Experimental procedures were approved by the local ethics committee (Comitato Etico Pediatrico Regionale Azienda Ospedaliero-Universitaria Meyer, Florence, Italy; protocol no. GR- 2013-02358262) and are in line with the declaration of Helsinki. All subjects gave written informed consent.

## 3. Results

### (a) Effects of adaptation on confidence and reaction times

We monitored decision confidence and reaction times (in an un-speeded task) while participants made numerosity judgements after adaptation, either to dense visual patterns or to hand-tapping. The major results were obtained from analysis of the 'aggregate observer', pooling data over all 12 participants (10 in the adaptation to hand-tapping). However, we also analysed individual data from all participants separately and, although the reduced data were necessarily more noisy, the group analysis gave essentially the same results as the

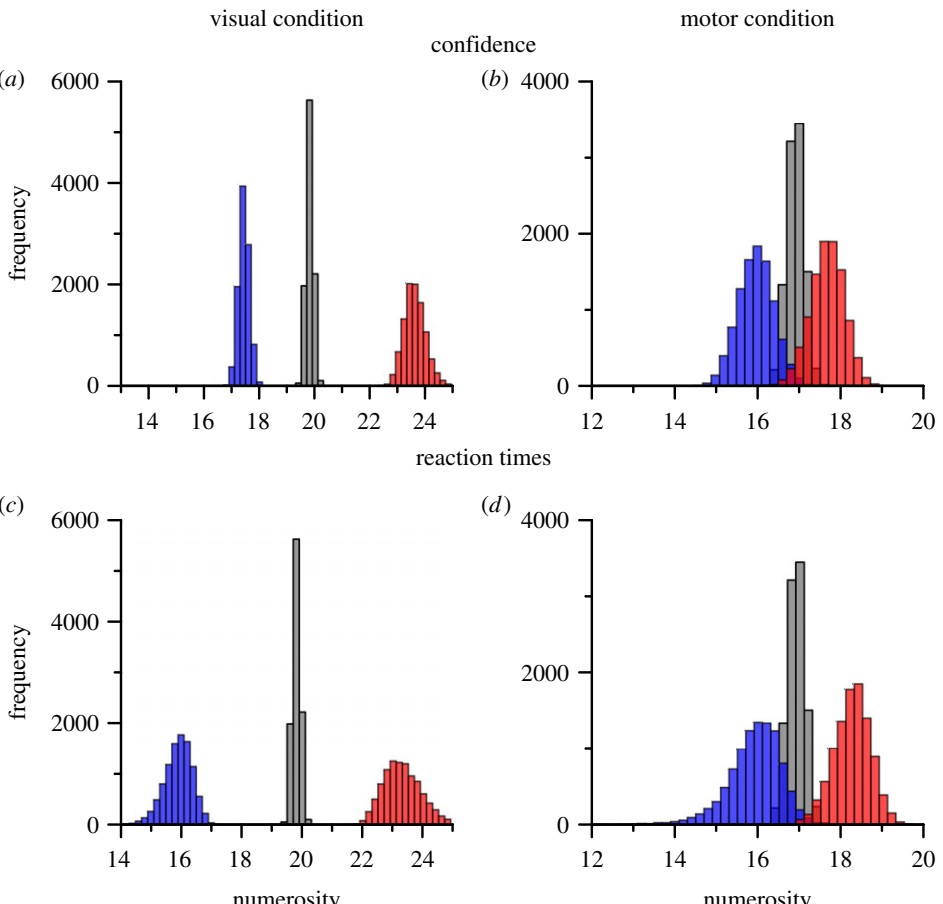

**Figure 4.** Frequency distributions of bootstraps for confidence (*a,b*) and reaction times (*c,d*), for visual or motor adaptation experiment (left and right panels, respectively). Data in blue represent visual baseline or slow tapping condition and red for high visual adaptation or fast motor tapping. Grey distributions show the bootstrapped mid-points between baseline (or slow) and adaptation (or fast tapping) PSEs. (Online version in colour.)

aggregate. The results of the individual analyses are reported in the electronic supplementary material, and summarized in figure S4 and table S1.

Figure 3 shows the main results from the aggregate data. Figure 3*a,b* indicate psychometric functions, plotting the proportion of trials (for all participants) where the test was reported as more numerous than the reference, as a function of the numerosity of the test patch. Both datasets were well fit by cumulative Gaussian error functions, which were clearly displaced by adaptation, both by visual dot-patterns and hand-tapping. In the unadapted condition (figure 3*a*, blue symbols and curves), the psychometric function was centred at 17 dots, very near the actual reference of 16 dots. Visual adaptation to 60 dots clearly displaced the psychometric function rightwards, shifting the median (which estimates the PSE) to 22.7 dots, meaning that after adaptation the probe needed to be 33% more numerous than the reference to appear equal to it. A similar effect occurred for hand-tapping: slow tapping had little effect, with the PSE remaining at 15.9 (near the reference), while fast tapping increased it to 18.1, again implying a decrease of apparent numerosity, in this case of 14%.

Both the confidence and mean reaction-time data were well fit by Gaussian functions ($R^2 > 0.75$ in all cases). The peaks of these functions (indicated by the arrows, and reported in the electronic supplementary material, table S1), clearly also shift with adaptation, both to visual numerosity and hand-tapping. The shift is in the same direction as the shift in PSEs, tending to align peaks in confidence and reaction times with the PSEs.

These results on the aggregate observer are very similar to those obtained from analysis of individual participants (see the electronic supplementary material).

The blue and red histograms of figure 4*a–d* show the results of bootstrapping (10 000 repetitions, sampling with replacement). On each repetition, estimates were made for PSE, point of *minimal confidence* and *maximal reaction time*. It is clear from inspection that in all cases the distributions for the investigated conditions overlap very little, indicating that they are significantly different. Bootstrap sign test yielded significance levels of $p < 0.003$ in all cases. On adaptation to visual stimuli peaks in both the confidence (figure 4*a*) and reaction time (figure 4*c*) were higher for the adapt-high condition than baseline in all 10 000 iterations ($p < 10^{-4}$). On adaptation to tapping, peaks in confidence (figure 4*b*) were lower for the adapt-high than adapt-low condition on only 34 iteration ($p = 0.0034$), and for reaction times (figure 4*d*) only 20 times ($p = 0.002$) out of 10 000.

We then used the bootstrapped distributions to pit two plausible models against each other: (i) that the shifts in the psychometric functions result from a response strategy for uncertain trials ([13]: illustrated in figure 1); (ii) that the change reflects adaptation-induced changes within sensory circuits. Model (i) predicts that the confidence and reaction-time distributions should not move with adaptation, so those for the adapt-high should be closer to $PSE_{base}$ (or $PSE_{low}$) than to $PSE_{high}$. On the other hand, model (ii) predicts that both peaks should follow the shifts in PSE, and therefore be closer

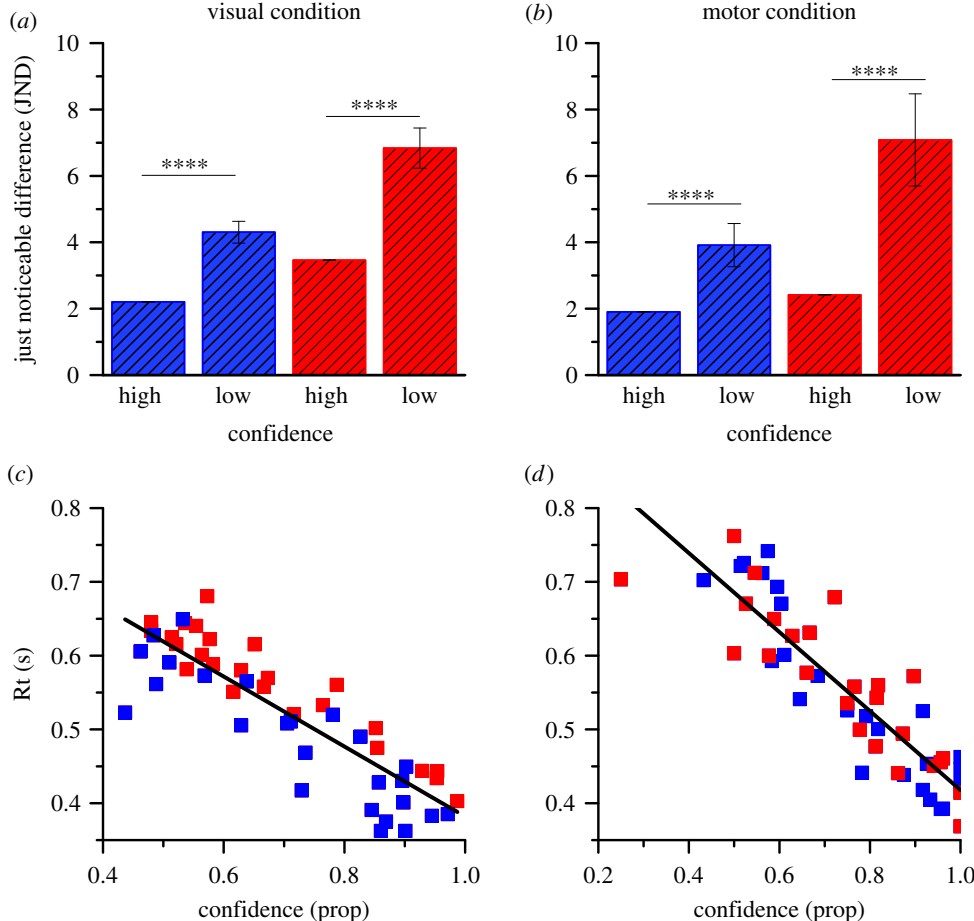

**Figure 5.** Bar graphs show precision for numerosity discrimination in the high or low confidence trials. In blue, data for baseline (or slow tapping) and red data for adaptation to high (or fast tapping) for visual and motor adaptation. (c,d) Reaction times (Rt; averaged over trials and subjects) as a function of confidence (averaged over trials and subjects) for the two adaptation conditions. Black lines represent the best-fitting linear regressions ((c) visual adaptation: $R^2 = 0.76$; (d) motor adaptation: $R^2 = 0.79$). Error bar represents $\pm 1$ s.e.m., ****$p < 0.0001$. (Online version in colour.)

to $PSE_{high}$. We tested this by bootstrap sign test, counting how many iterations were closer to $PSE_{base}$ (or $PSE_{low}$) than $PSE_{high}$. We also bootstrapped the PSEs themselves on each iteration, to include their error in the calculation (the grey distribution in figure 4 shows the bootstrapped mid-points of the two PSEs). For visual adaptation, not a single iteration of either confidence or reaction-time peaks was closer to $PSE_{base}$ than $PSE_{high}$, implying the likelihood for the first model is $p < 10^{-4}$. The tapping condition also showed a clear effect. For the confidence data, the likelihood of model (i) was $p = 0.05$, compared with $p = 0.95$ for model (ii), giving a likelihood ratio of 19. Reaction times were more significant, with likelihood of model (i) equal to 0.0064 compared with 0.9936 for model (ii), 166 times less likely. All the bootstrapped sign tests provide strong evidence for model (ii) for both types of adaptation, suggesting that the adaptation occurs within sensory rather than decision systems.

To test the validity of the confidence ratings, we separated the data into high- and low-confidence trials and fitted psychometric functions separately for each, calculating the JND, from the standard deviation of the fit. Standard errors and significance were calculated by bootstrap. As there were three times as many trials judged confident than unconfident, the data for confident judgements were under sampled during bootstrapping to match sample sizes. Figure 5a,b shows JNDs for the high-confidence trials were significantly lower than that for low-confidence, by at least a factor of two ($p <$

$10^{-4}$ in all cases), consistent with the idea that subjective confidence reflects a genuine metacognitive ability which assesses the quality of sensory evidence [16].

We also correlated reaction times against confidence (figure 5c,d). Each point of figure 5c comes from figure 3c,e, and those from figure 5d from figurs 3d,f. The correlation was strong, with $r = -0.87$ and $-0.89$ for the two adaptation types, accounting for more than 70% of the variance. This shows that the two measures covary together, consistent with their being driven by a common factor, most probably perceived stimulus strength.

## (b) Control experiment: effects of reward on confidence and reaction times

In order to show that confidence and reaction times do not necessarily change with PSE, we devised a control experiment where we manipulated rewards. Here, there was no adaptation, but participants played a point-based game, with three types of reward regimes (in different blocks). In baseline blocks, they received 1 point for each correct response and lost 1 for every error (performing on average at 85% correct). In 'reward-low' blocks, they received 2 points for correctly responding 'less than', and lost 1 each error; and in 'reward-high', 2 points for correctly responding 'greater than', losing 1 for an error. This simple reward manipulation of rewards biased observers towards the

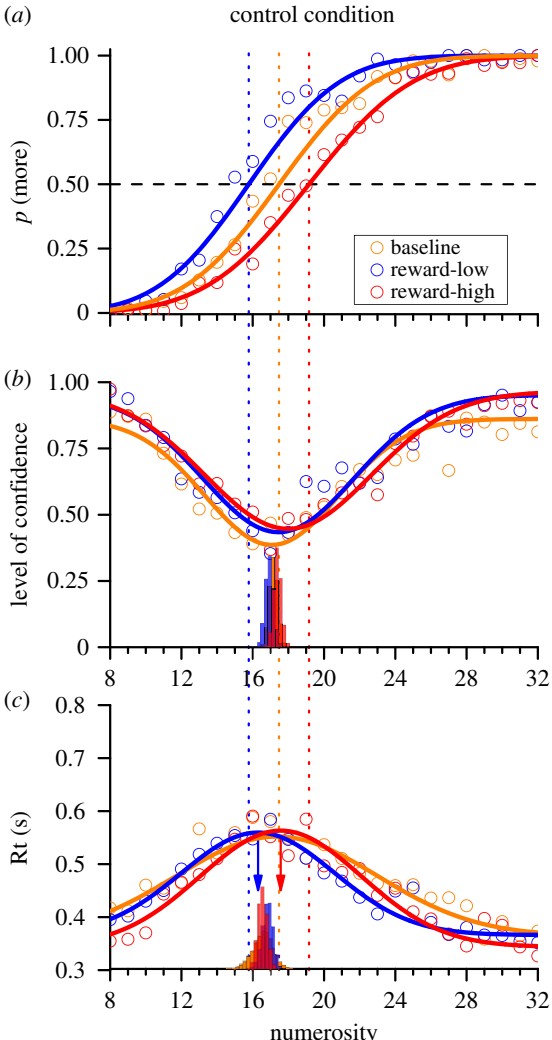

**Figure 6.** (a) Psychophysical functions of proportion of trials when the test was seen as more numerous than the neutral probe, as a function of physical numerosity in the control condition (baseline in orange, leftward condition in blue and rightward condition in red). (b) Expressions of confidence, as a function of physical numerosity. (c) Mean reaction times (Rt; in seconds) as a function of physical numerosity. The continuous dotted lines indicate the PSE of the psychophysical curves. The histograms below the confidence and reaction time fits represent the bootstrap analysis. (Online version in colour.)

how many iterations were nearer to the PSE of that condition rather than to the PSE of the baseline (non-rewarded) condition. For the confidence measures, the results were clear: the probabilities of model (ii) (closer to the shifted PSE) being correct were $p = 0.046$ for the reward-low condition, and $p = 10^{-4}$ for the reward-high condition, 20 and 10 000 times less likely than model (i). The results for reaction times was similarly in favour of model (i), with probabilities for model (ii) at $p < 10^{-4}$ for the reward-low condition, and $p = 0.012$ for the reward-high condition, infinite and 81 times less likely than model (i). Reaction times in this experiment may have been less reliable, because of variable slowing when integrating the reward *prior*. Again, the results from the aggregate observer are very similar to those obtained from analysis of individual participants (see the electronic supplementary material).

## 4. Discussion

The primary goal of this study was to probe the mechanisms of numerosity adaptation, to test whether adaptation affects sensory processing mechanisms directly, or indirectly via decision or response criteria. We argue that a change in sensory processing should result in a comparable change in minimum decision confidence and maximum reaction times, which should shift to align with the point of subjective equality after adaptation, where the test and probe stimuli are, by definition, most similar perceptually. On the other hand, if the change in PSE results from a response bias, the peaks in confidence and reaction times should not change with adaption (figure 1). Our results clearly support the claim that adaptation affects sensory processing directly. Two types of adaptation—to visual patterns and to hand-tapping—caused large shifts in PSEs, with concomitant shifts in peak confidence and reaction times. In all cases, the sensory processing model was far more probable than that suggested by confidence-induced shifts in response criteria. On the other hand, when the PSEs were shifted by awarding rewards for specific responses, the shifts in PSE were not accompanied by shifts in confidence or reaction times.

The results are interesting for several reasons. Firstly, there has been a long-standing debate about the nature of numerosity processing, particularly about whether it is sensed directly, or is a by-product of texture processing [17,18]. One of the strongest lines of evidence that numerosity is distinct from texture density comes from adaptation studies, particularly cross-modal and cross-format adaptation [7]: adapting to sequences of flashes or tones affects the perceived numerosity of dot arrays, difficult to ascribe to texture perception. The demonstration that adaptation to fast or slow hand-tapping changes the perceived numerosity of spatial arrays is even more fascinating, as it links perception and action, implicating common mechanisms for perceiving and reproducing numerosity [8].

However, as first argued by Laplace [19], extraordinary claims require extraordinary evidence. It is, therefore, reasonable to expect a rigorous demonstration that motor tapping affects the perception of numerosity directly, rather than merely biasing the decision or the response along the lines of figure 1. The fact that all analyses show that both confidence and reaction-time peaks move to the adapted PSE strongly favours the hypothesis that adaptation causes

double-reward response when uncertain, causing robust shifts in the PSE. Figure 6a shows the psychometric functions for the aggregate observer for the three conditions. The PSE for the standard condition was 17.5 (a constant bias of 1.5 from the physical equivalent of 16), while for the 'reward-low' condition it was 15.8 (1.7 lower) and for 'reward-high' was 19.1 (1.6 higher). Both cases are near the predictions of the ideal observer (which predicts a shift of 1.2 towards the rewarded side).

However, the shift in PSE was not accompanied by concomitant shifts in confidence: the minima in the Gaussians are very similar for all three conditions (17.4, 17.1 and 18.0 for low, baseline and high). Similarly, the peak reaction times did not follow the PSEs, but again tended to cluster around the baseline PSE (16.3, 17.3 and 17.6). The histograms below the confidence and reaction-time curves show the bootstrap analysis, similar to that of figure 4. The bootstraps clearly overlap considerably. Again, we tested the two plausible models outlined for figure 4, counting, for each condition,

changes at the sensory level. This has important ramifications for understanding the role of numerosity mechanisms in perception and action, relating well to the electrophysiological studies showing a clear selectivity for the number of self-produced actions in area 5 of the superior parietal lobule of the monkey [10,20].

The other more general result of this study is a method of validating adaptation and other effects of temporal and spatial dependency (such as serial dependence [21–24]). Adaptation is a fundamental tool in psychophysics, famously referred to as 'the psychophysicist's microelectrode' [11]. However, adaptation studies necessarily rely on subjective judgements, on participants reporting their subjective impressions. Most modern adaptation studies use two-alternative forced choice techniques that ask participants to compare the adapted test to a probe, yielding psychometric functions from which the point of subjective equality can be titrated. However, unlike other forced choice tasks (such as measurement of contrast sensitivity), there is no right or wrong answer: just a subjective judgement that stimulus A was larger, brighter or more numerous than stimulus B. Over a considerable range around the point of subjective equality, judgements are difficult, but participants must respond, guessing if unsure. It requires only a slight tendency to respond stereotypically in one direction when unsure to shift the curves, robustly changing the PSE, without changing the slope of the function [13]. It, therefore, becomes important to have objective corroborative evidence that the point of subjective equality really reflects sensory changes rather than response biases. Gallagher et al. [14] suggested that minima in response criteria could provide useful corroboration, and demonstrated that it can do so for motion adaption (and also for serial dependence). We extend their idea, showing that even with a far more subtle form of adaptation elicited by hand-tapping, the minima in confidence follow the changes in PSE.

We point out that we are testing a specific model of how decision criteria may affect PSEs: that a small tendency of response bias could affect trials of low confidence, causing reliable shifts in PSE [13]. With this particular model, as confidence is driving the response, it is unlikely to shift with the response PSE. However, other more complex models of perceptual decisions [25,26] may predict that confidence and reaction times do change with changes in PSE. Indeed, with these classes of models it is often difficult to distinguish experimentally between sensory and perceptual decision effects [27]. We, therefore, designed a realistic experiment that manipulated PSEs at the decisional level, by rewarding correct responses in a specific direction (high or low). This produced robust changes in responses, shifting the PSE as expected, as participants sought to optimize gains: however, the shifts in PSE were not accompanied by concomitant changes in confidence, nor in reaction times. This is a clear existence proof that at least some types of manipulation on decisions are not paralleled by shifts in confidence, which may, therefore, be a signature of sensory changes. Gallagher et al. [14] performed a similar experiment, instructing participants specifically to respond 'left' or 'right' when confidence is low, and also showed that this manipulation does not shift the point of minimal confidence. However, our task was more natural, in that we gave no instructions to participants on how to respond, nor that they should take confidence into account. It was a natural task with greater risks on one side

than the other (like those pioneered by Trommershäuser et al. [28]) which human participants soon learn to optimize. Yet this very natural and spontaneous task, which shifted PSEs smoothly, caused no similar shifts in confidence or reaction times.

In general, reaction times provided more robust data than confidence for the sensory shifts in PSE. Reaction times could have several advantages to confidence measures. Firstly, they are objective and come at no extra cost, automatically encoded in the timestamps of the stimuli and responses, without having to ask participants to make a second response. Nor was it necessary to ask for a speeded response; we simply relied on the tendency of participants to respond reasonably quickly in order to finish the session as soon as possible. For the adaptation experiments, reaction times proved to be more informative than confidence, in all cases providing stronger evidence for a shift in their peak. For example, for the aggregate data for adaptation to tapping, the log Bayes factor ($\log_{10}BF_{12}$) was 1.26 for confidence, compared with 2.22 for reaction-time data. For the analysis of individual data (where there are far fewer trials, hence more noisy estimates) the $\log_{10}BF_{12}$ for confidence was 1.14 compared with 2.46 for reaction times. In all cases, the $\log_{10}$-Bayes factors were greater than 1, considered *strong* evidence, but the reaction-time data gave $\log_{10}BF > 2$, considered *decisive* [29]. There is considerable evidence showing that reaction times vary monotonically with signal strength [15], and should, therefore, be maximal at the point of least difference in the signals. Combined with the ease with which reaction-time data can be collected, with no additional load on participants, it would appear to be the preferred method.

To summarize, we present a new technique for investigating the mechanisms of numerosity adaptation and sensory adaptation in general. By simultaneously measuring subjective confidence and more importantly—reaction times, we demonstrate that adaptation to numerosity, either by observing visual stimuli of high numerosity or by subjects tapping in a particular region occurs at a sensory level, before stages of perceptual decision. Adaptation affects not only perceived numerosity, but also subjective confidence and reaction times, showing that they are a consequence of sensory adaptation, rather than the cause for the shift in the psychometric functions.

**Ethics.** Experimental procedures were approved by the local ethics committee (Comitato Etico Pediatrico Regionale Azienda Ospedaliero-Universitaria Meyer, Florence, Italy; protocol no. GR-2013-02358262) and are in line with the declaration of Helsinki. All subjects gave written informed consent.

**Data accessibility.** The data that support the findings of this study are available online from the Dryad Digital Repository: https://doi.org/10.5061/dryad.95x69p8gh [30].

**Authors' contributions.** All authors participated in conceiving and designing the experiments, discussion of data and writing the manuscript. P.A.M.M. ran all the experiments. P.A.M.M., G.M.C. and D.C.B. analysed the data.

**Competing interests.** We declare we have no competing interests.

**Acknowledgements.** This research was funded from the European Union (EU) and Horizon 2020—grant agreement no. 832813—ERC Advanced 'Spatio-temporal mechanisms of generative perception—GenPercept'; from the Italian Ministry of Education, University, and Research under the PRIN2017 programme (grant no. 2017XBJN4F—'Environ-Mag' and grant no. 2017SBCPZY—'Temporal context in perception: serial dependence and rhythmic oscillations').

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
