## [Reviewer comments · Proceedings of the Royal Society B: Biological Sciences]

Review History

RSPB-2020-0139.R0 (Original submission)

Review form: Reviewer 1

Recommendation

Major revision is needed (please make suggestions in comments)

Scientific importance: Is the manuscript an original and important contribution to its field?

Good

General interest: Is the paper of sufficient general interest?

Good

Quality of the paper: Is the overall quality of the paper suitable?

Good

Is the length of the paper justified?

Yes

Should the paper be seen by a specialist statistical reviewer?

No

Do you have any concerns about statistical analyses in this paper? If so, please specify them explicitly in your report.

Yes

It is a condition of publication that authors make their supporting data, code and materials available - either as supplementary material or hosted in an external repository. Please rate, if applicable, the supporting data on the following criteria.

Is it accessible?

No

Is it clear?

No

Is it adequate?

No

Do you have any ethical concerns with this paper?

No

Comments to the Author

Please see attached file. (See Appendix A)

Review form: Reviewer 2

Recommendation

Accept with minor revision (please list in comments)

Scientific importance: Is the manuscript an original and important contribution to its field?

Good

General interest: Is the paper of sufficient general interest?

Good

Quality of the paper: Is the overall quality of the paper suitable?

Good

Is the length of the paper justified?

Yes

Should the paper be seen by a specialist statistical reviewer?

Yes

Do you have any concerns about statistical analyses in this paper? If so, please specify them explicitly in your report.

Yes

It is a condition of publication that authors make their supporting data, code and materials available - either as supplementary material or hosted in an external repository. Please rate, if applicable, the supporting data on the following criteria.

Is it accessible?

N/A

Is it clear?

N/A

Is it adequate?

N/A

Do you have any ethical concerns with this paper?

No

Comments to the Author

General

In the manuscript entitled "adaptation to hand-tapping affects sensory processing of numerosity directly: evidence from reaction time and confidence" tested whether the adaptation effect of numerosity may reflect just response biases or the modification of sensory process. This manuscript presented a detailed function of visual and motor adaptation for quantity representation process. The data collected in this study contribute to our understanding how effect adaptation on the numerical processing. However, I suspect that the sample size seems to be small to evaluate these results. The manuscript could be considered for publication following revisions.

#1

I think it is better to attach the fitting plot from each participant as an appendix. Because of the very small sample size (especially, only 6 participants for visual adaptation condition), it's important to show that the data of adaptation results of each participant to understand the effect size.

#2

The major concern is a small sample size. At least, The sample size estimation and power analysis are needed for both conditions (visual adaptation experiment and tapping experiment). I recommend to enlarge sample size.

#3

Please indicate the number of trials for each participant performed actually for each condition.

Decision letter (RSPB-2020-0139.R0)

24-Feb-2020

Dear Professor Burr:

I am writing to inform you that your manuscript RSPB-2020-0139 entitled "Adaptation to hand-tapping affects sensory processing of numerosity directly: evidence from reaction times and confidence" has, in its current form, been rejected for publication in Proceedings B.

This action has been taken on the advice of referees, who have recommended that substantial revisions are necessary. With this in mind we would be happy to consider a resubmission,

provided the comments of the referees are fully addressed. However please note that this is not a provisional acceptance.

Please find below the comments made by the referees, not including confidential reports to the Editor, which I hope you will find useful. Please note in particular the extensive treatment by reviewer 1, who has identified a substantive problem which will require careful consideration. Also please consider carefully the issue of sample size, as raised by reviewer 2. If you do choose to resubmit your manuscript, please upload the following:

Sincerely,
 Dr Robert Barton
 mailto: proceedingsb@royalsociety.org

Associate Editor
 Board Member: 1
 Comments to Author:

Your manuscript has now been seen by reviewers. They have raised some significant concerns and I am afraid that on the basis of those concerns I must recommend rejection. Nevertheless, I am also recommending that you be given an opportunity to deal with them -- particularly those of reviewer 1.

Reviewer(s)' Comments to Author:

Referee: 1

Comments to the Author(s)
 Please see attached file.

Referee: 2

Comments to the Author(s)
 General

In the manuscript entitled "adaptation to hand-tapping affects sensory processing of numerosity directly: evidence from reaction time and confidence" tested whether the adaptation effect of numerosity may reflect just response biases or the modification of sensory process. This manuscript presented a detailed function of visual and motor adaptation for quantity representation process. The data collected in this study contribute to our understanding how

effect adaptation on the numerical processing. However, I suspect that the sample size seems to be small to evaluate these results. The manuscript could be considered for publication following revisions.

#1

I think it is better to attach the fitting plot from each participant as an appendix. Because of the very small sample size (especially, only 6 participants for visual adaptation condition), it's important to show that the data of adaptation results of each participant to understand the effect size.

#2

The major concern is a small sample size. At least, The sample size estimation and power analysis are needed for both conditions (visual adaptation experiment and tapping experiment). I recommend to enlarge sample size.

#3

Please indicate the number of trials for each participant performed actually for each condition.

Author's Response to Decision Letter for (RSPB-2020-0139.R0)

See Appendix B.

RSPB-2020-0801.R0

Review form: Reviewer 1

Recommendation

Accept as is

Scientific importance: Is the manuscript an original and important contribution to its field?

Good

General interest: Is the paper of sufficient general interest?

Good

Quality of the paper: Is the overall quality of the paper suitable?

Excellent

Is the length of the paper justified?

Yes

Should the paper be seen by a specialist statistical reviewer?

No

Do you have any concerns about statistical analyses in this paper? If so, please specify them explicitly in your report.

No

It is a condition of publication that authors make their supporting data, code and materials available - either as supplementary material or hosted in an external repository. Please rate, if applicable, the supporting data on the following criteria.

Is it accessible?

N/A

Is it clear?

N/A

Is it adequate?

N/A

Do you have any ethical concerns with this paper?

No

Comments to the Author

The authors should be congratulated on their thorough response to the reviewers' comments. The authors have addressed my main concern with the control experiment. The manuscript as a whole is greatly improved from the original version, both in terms of strength and read-ability. I can recommend that the paper be accepted for publication in its current form.

I could recommend one further improvement, which the authors can choose not to follow. Currently the analysis shows no significant shift in confidence with the shift in PSE due to reward contingencies in the control experiment. It is clear from Figure 6B that this is indeed robust. However, rather than arguing for no effect, the authors could show that the relative shift in confidence (and reaction time) with PSE for adaptation (a relative shift of about 1) is significantly greater than the relative shift with reward contingencies. The authors could then argue for significant evidence against the null hypothesis that confidence and reaction times are shifted by decision strategies as opposed to a genuine perceptual effect of adaptation. In saying this, I would happily accept the manuscript in its current form without this additional analysis.

Supplementary materials, Figure S3, subject S04, PSE: there is a blue line which I believe should be red.

Decision letter (RSPB-2020-0801.R0)

28-Apr-2020

Dear Professor Burr

I am pleased to inform you that your manuscript RSPB-2020-0801 entitled "Adaptation to hand-tapping affects sensory processing of numerosity directly: evidence from reaction times and confidence" has been accepted for publication in Proceedings B.

The referee has recommended publication, but has suggested a minor revision to your manuscript. Therefore, I invite you to revise your manuscript. Because the schedule for publication is very tight, it is a condition of publication that you submit the revised version of your manuscript within 7 days. If you do not think you will be able to meet this date please let us know.

To revise your manuscript, log into <https://mc.manuscriptcentral.com/prsb> and enter your Author Centre, where you will find your manuscript title listed under "Manuscripts with

Decisions." Under "Actions," click on "Create a Revision." Your manuscript number has been appended to denote a revision. You will be unable to make your revisions on the originally submitted version of the manuscript. Instead, revise your manuscript and upload a new version through your Author Centre.

[http://datadryad.org/submit?journalID=RSPB&manu=\(Document not available\)](http://datadryad.org/submit?journalID=RSPB&manu=(Document+not+available)) which will take you to your unique entry in the Dryad repository. If you have already submitted your data to dryad you can make any necessary revisions to your dataset by following the above link.

Please see <https://royalsociety.org/journals/ethics-policies/data-sharing-mining/> for more details.

Sincerely,

Dr Robert Barton

Associate Editor

Board Member

Comments to Author:

I am delighted to recommend publication of your manuscript in Proceedings. You have done an excellent job in responding to the reviewers' comments. Congratulations. You may wish to consider the suggestion that the reviewer has made about an additional analysis. But I leave that to you. [Note that there may be an error in Fig. S3.]

Reviewer(s)' Comments to Author:

Referee: 1

Comments to the Author(s).

The authors should be congratulated on their thorough response to the reviewers' comments. The authors have addressed my main concern with the control experiment. The manuscript as a whole is greatly improved from the original version, both in terms of strength and read-ability. I can recommend that the paper be accepted for publication in its current form.

I could recommend one further improvement, which the authors can choose not to follow.

Currently the analysis shows no significant shift in confidence with the shift in PSE due to reward contingencies in the control experiment. It is clear from Figure 6B that this is indeed robust.

However, rather than arguing for no effect, the authors could show that the relative shift in confidence (and reaction time) with PSE for adaptation (a relative shift of about 1) is significantly greater than the relative shift with reward contingencies. The authors could then argue for significant evidence against the null hypothesis that confidence and reaction times are shifted by decision strategies as opposed to a genuine perceptual effect of adaptation. In saying this, I would happily accept the manuscript in its current form without this additional analysis.

Supplementary materials, Figure S3, subject S04, PSE: there is a blue line which I believe should be red.

Author's Response to Decision Letter for (RSPB-2020-0801.R0)

See Appendix C.

Decision letter (RSPB-2020-0801.R1)

29-Apr-2020

Dear Professor Burr

I am pleased to inform you that your manuscript entitled "Adaptation to hand-tapping affects sensory processing of numerosity directly: evidence from reaction times and confidence" has been accepted for publication in Proceedings B.

Your article has been estimated as being 9 pages long. Our Production Office will be able to confirm the exact length at proof stage.

Open Access

You are invited to opt for Open Access, making your freely available to all as soon as it is ready for publication under a CCBY licence. Our article processing charge for Open Access is £1700. Corresponding authors from member institutions (<http://royalsocietypublishing.org/site/librarians/allmembers.xhtml>) receive a 25% discount to these charges. For more information please visit <http://royalsocietypublishing.org/open-access>.

Paper charges

Sincerely,
Proceedings B
<mailto:proceedingsb@royalsociety.org>

Appendix A

In this manuscript the authors analyse the effect of numerosity adaptation on perceptual decisions, confidence, and reaction times. The authors present a clear argument that if numerosity adaptation is indeed a sensory effect, then the distribution of confidence and reaction times should be shifted (to the same extent as the PSE). The data indeed show shifted distributions of confidence ratings and reaction times, and the authors conclude that numerosity adaptation must cause a shift in the perceptual representation of numerosity. Although I found the manuscript generally convincing and of sufficient scientific rigor for publication, I worry that the authors' null hypothesis is inappropriate. I outline this further below, in addition to some further recommendations for improvement.

The authors show evidence for the alternative hypothesis, that confidence and reaction time distributions are shifted with adaptation, against the null hypothesis that there is no shift. They conclude that this is evidence that visual and motor adaptation to numerosity results in a sensory shift in the representation of numerosity, as opposed to a shift in decision strategies. The null hypothesis therefore assumes that a shift in decision strategies will leave confidence and reaction time distributions unchanged. However, standard models do suggest changes in confidence and reaction time with changes in decision strategies. Confidence is often modelled as placing additional confidence criteria *relative* to the perceptual decision criterion (for example, see **Maniscalco and Lau, 2014**, Figure 3.3a). In this case, confidence would also shift with a biased decision criterion. Diffusion models of choice and reaction times suggest reaction times can also change as a result of the 'starting point' – which is thought to be equivalent to a response bias (see for example, **Ratcliff and McKoon, 2008**). The true null hypothesis for concluding that numerosity adaptation induces sensory shifts as opposed to a shift in response bias, is a null model that incorporates the expected shifts in confidence and reaction times to changes in response bias. The current analysis may not be sensitive enough to distinguish these two models. I have run some simulations to explain this point more clearly (I realise this is a bit over-the-top for a review, but I feel this is important):

Shifts in confidence resulting from two models of the effect of numerosity adaptation. The left column demonstrates the alternative hypothesis: a sensory shift in numerosity perception. The right column demonstrates the null hypothesis: a change in response bias. **Top left:** solid green distributions represent the distribution of signal strength for five test stimuli (8:5:28 dots) minus the probe stimulus (16 dots). An unbiased observer places their perceptual response criterion at 0 (responding 'Right' when the test/right stimulus appears more numerous than the probe; black vertical line). The red vertical lines show the confidence criteria, where the observer responds 'low confidence' when the difference in signal strength is between these two criteria. Sensory adaptation shifts the representation of numerosity – the test stimuli appear less numerous – shown by the distributions with dotted lines of corresponding colours. **Bottom left:** proportion of 'Right' responses (solid) and proportion high confident (dotted) for baseline (blue) and post-adaptation (red), as shown in Figure 3A and 3C in the manuscript,

but generated by simulating the model shown above. **Top right:** The solid lines are the same as in the top left plot. In this model adaptation causes a shift in the perceptual response criterion – the observer requires more evidence to respond ‘Right’ – shown in the dotted black vertical line. The confidence criteria technically remain the same: the observer requires the sensory evidence to be the same amount stronger/weaker than their perceptual decision criterion in order to give a high confidence rating. However, this means a shift in the position of the confidence criteria on the x-axis (red vertical dotted lines). **Bottom right:** same as for bottom left, however the red curves are generated by the response bias model as opposed to the sensory adaptation model.

Shifts in reaction times resulting from two models of the effect of numerosity adaptation. The left column demonstrates the alternative hypothesis: a sensory shift in numerosity perception. The right column demonstrates the null hypothesis: a change in response bias. **Top left:** Thick solid curves represent histograms of reaction times to stimuli of varying drift rates (numerosity; the mean drift rate is shown in thin lines) for ‘Left’ (lower threshold) and ‘Right’ (upper threshold) responses (scaled for visibility). The sensory adaptation hypothesis suggests a shift in the drift rate – the adapted stimuli now have stronger evidence for lower numerosity – shown in the dotted thin lines. This results in a shift in the reaction times (dotted distributions). **Bottom left:** Proportion ‘right’ responses (solid) and distribution of reaction times (dotted) for baseline (blue) and post-adaptation (red) generated from the above diffusion model. A non-decision time of 300 ms was added to reaction times to make them more realistic. **Top right:** Solid lines show the same as the top left. Dotted lines show the drift rates and reaction time histograms after a change in the starting point (a response bias: the observer requires less evidence to respond ‘Left’). **Bottom right:** Same as bottom left, but the red curves are generated from simulations of the response bias model.

It is unfortunate, but the above simulations demonstrate that response bias could equally account for the authors’ data in its current form. The authors therefore do not present the correct hypothesis test to conclude that numerosity adaptation shifts the perceptual representation of numerosity as opposed to shifting the decision criterion. A more sensitive analysis may be able to distinguish between the sensory and response bias hypotheses with the current data.

Further recommendations:

The analysis using the ANOVA on individual subjects could be removed. I suggest this for two reasons: first, the authors conclude from this analysis that “All this suggests that adaptation affected PSE, confidence, and reaction-times by a similar amount”, when in fact the analysis showed that there was no significant effect of ‘index’ (a

critical reader could argue that the lack of significant difference is due to the experiment being under-powered for this test). Second, this conclusion is supported by the findings of the previous and following analyses, so the ANOVA itself is unnecessary.

The strongest results are based on data where the trials of individual participants are aggregated. The reader would feel more confident that this approach is warranted if there were some mention of how similar the effect was across participants, prior to presenting the aggregated data.

Minor points:

1. There are some missing details in the methods:
 - a. Number of trials per participant (or condition)
 - b. Clarify the response keys: line 118 suggest left/right arrows for confidence, line 135 suggests up/down arrows
2. Is there a reason why the points on figure 3A do not have error bars?
3. Is the gaussian fitted to mean or median reaction times?
4. Line 236, the term 'were higher' is a little ambiguous. If I have understood correctly, the authors mean that peaks (means?) in the distributions of confidence and reaction times corresponded to greater numerosity.
5. Was bootstrapping performed over trials irrespective of participant, or over participants, or over trials but equally within participants?

References:

Maniscalco, B., & Lau, H. (2014). Signal detection theory analysis of type 1 and type 2 data: meta- d' , response-specific meta- d' , and the unequal variance SDT model. In *The cognitive neuroscience of metacognition* (pp. 25-66). Springer, Berlin, Heidelberg.

Ratcliff, R., & McKoon, G. (2008). The diffusion decision model: theory and data for two-choice decision tasks. *Neural computation*, 20(4), 873-922.

Appendix B

Dear Dr Barton and referees

Thank you for the opportunity to revise the manuscript. We are very grateful to the referees for their conscientious effort, particularly referee 1 who went well beyond the call of duty. We took all their comments very seriously, responding not just with a revision, but much more data.

Below we treat in detail all the referees' comments, but I'd first make a general point. Referee 1 quite correctly points out that some models of perceptual decision do predict shifts in confidence and RTs. Indeed, historically, decision models can predict many effects assumed to be sensory, including properties as basic as the steepness of the psychometric function and the dipper function (Denis Pelli's work of the 80s). Often, as in this case, the predictions are too close to discriminate.

So we tackled the challenge head-on with a new experiment where we biased responses with selective incentives. This shifted the point of subjective equality towards the rewarded response, but the shift was not accompanied by changes in minimal confidence or maximal RTs. This shows that changing responses in a way that should not affect primary sensory processing does not shift confidence or RTs, as does adaptation. We have made space for the new data by consigning the individual data to supplemental material (consistent with advice from both referees).

We also increased the number of participants in all experiments to reinforce our results. We had intended to add even more participants, but have been in lock-down for a month. However, with the added new experiment, and extra data on the original tasks, we believe our results are now really robust, and ready for publication in the proceedings.

Yours sincerely

David Burr

Specific reply to referees

Referee 1

In this manuscript the authors analyse the effect of numerosity adaptation on perceptual decisions, confidence, and reaction times. The authors present a clear argument that if numerosity adaptation is indeed a sensory effect, then the distribution of confidence and reaction times should be shifted (to the same extent as the PSE). The data indeed show shifted distributions of confidence ratings and reaction times, and the authors conclude that numerosity adaptation must cause a shift in the perceptual representation of numerosity. Although I found the manuscript generally convincing and of sufficient scientific rigor for publication, I worry that the authors' null hypothesis is inappropriate. I outline this further below, in addition to some further recommendations for improvement.

Thank you

The authors show evidence for the alternative hypothesis, that confidence and reaction time distributions are shifted with adaptation, against the null hypothesis that there is no shift. They conclude that this is evidence that visual and motor adaptation to numerosity results in a sensory shift in the representation of numerosity, as opposed to a shift in decision strategies. The null hypothesis therefore assumes that a shift in decision strategies will leave confidence and reaction

time distributions unchanged. However, standard models do suggest changes in confidence and reaction time with changes in decision strategies. Confidence is often modelled as placing additional confidence criteria *relative* to the perceptual decision criterion (for example, see **Maniscalco and Lau, 2014**, Figure 3.3a). In this case, confidence would also shift with a biased decision criterion. Diffusion models of choice and reaction times suggest reaction times can also change as a result of the ‘starting point’ – which is thought to be equivalent to a response bias (see for example, **Ratcliff and McKoon, 2008**). The true null hypothesis for concluding that numerosity adaptation induces sensory shifts as opposed to a shift in response bias, is a null model that incorporates the expected shifts in confidence and reaction times to changes in response bias. The current analysis may not be sensitive enough to distinguish these two models. I have run some simulations to explain this point more clearly (I realise this is a bit over-the-top for a review, but I feel this is important):

It is unfortunate, but the above simulations demonstrate that response bias could equally account for the authors’ data in its current form. The authors therefore do not present the correct hypothesis test to conclude that numerosity adaptation shifts the perceptual representation of numerosity as opposed to shifting the decision criterion. A more sensitive analysis may be able to distinguish between the sensory and response bias hypotheses with the current data.

All this is very important, which we now discuss (briefly, with the space constraints). However, we make two points.

Firstly, we are testing the *specific* idea of Morgan and colleagues (2011) that the low confidence (together with a decision rule) can lead to substantial shifts in PSE. If the confidence is driving the decision, it cannot move with the decision. We do, however, accept that models of decision can almost perfectly emulate the effects of sensory processing, and discuss this point.

As many models of high-level processes can be effectively indistinguishable from low-level effects (clearest example is uncertainty compared with non-linear transducer to explain the dipper function), we thought it best to tackle the problem head-on experimentally, with a counterexample where shifts in PSE are not accompanied by shifts in confidence. Rather than replicate Morgan’s technique (which Arnold has also tested), where subjects were told to respond in a specific way if unsure, we devised a more natural task where we manipulated the reward structure, with double penalties for one side or the other. Subjects were not told how to deal with the penalties, but learned to avoid that side, which shifted PSEs (as it would for an ideal observer): but there were no concomitant changes in confidence or RTs. We believe this existence proof shows that the shifts in confidence are not a necessary consequence of decisional changes.

Nevertheless, your point is fundamental, and we are very grateful for having it pointed out, greatly strengthening the manuscript.

Further recommendations:

The analysis using the ANOVA on individual subjects could be removed. I suggest this for two reasons: first, the authors conclude from this analysis that “All this suggests that adaptation affected PSE, confidence, and reaction-times by a similar amount”, when in fact the analysis showed that there was no significant effect of ‘index’ (a critical reader could argue that the lack of significant difference is due to the experiment being under-powered for this test). Second, this conclusion is supported by the findings of the previous and following analyses, so the ANOVA itself is unnecessary.

Thank you for this advice, which we have taken. We now move all results from the group analysis of individual participants to supplemental material – together with the individual graphs – and remove the ANOVA. We point out that the group data are in the same direction as the aggregate. Removing this figure made space for the new experiment, so it worked out well.

The strongest results are based on data where the trials of individual participants are aggregated. The reader would feel more confident that this approach is warranted if there were some mention of how similar the effect was across participants, prior to presenting the aggregated data.

Again we agree. The first paragraph of the results section now makes this point, and the interested reader can check supplemental material at will. The raw data are also available.

Minor points:

1. There are some missing details in the methods: a. Number of trials per participant (or condition)

Thank you for that, we added the number of trials.

b. Clarify the response keys: line 118 suggest left/right arrows for confidence, line 135 suggests up/down arrows

Now we clarified the response keys.

2. Is there a reason why the points on figure 3A do not have error bars?

Well these are averages of binary data (0 or 1) – the errors would not be very useful. The same is true of the confidence ratings. So in the end, we dropped all error bars, they only added clutter to the graphs. This individual results are now in supplemental material for the reader, together with summary graphs and tables.

3. Is the gaussian fitted to mean or median reaction times?

We used the mean of the reaction times (now stated in the manuscript).

4. Line 236, the term 'were higher' is a little ambiguous. If I have understood correctly, the authors mean that peaks (means?) in the distributions of confidence and reaction times corresponded to greater numerosity.

5. Was bootstrapping performed over trials irrespective of participant, or over participants, or over trials but equally within participants?

Equally within participants.

Referee: 2

General

In the manuscript entitled "adaptation to hand-tapping affects sensory processing of numerosity directly: evidence from reaction time and confidence" tested whether the adaptation effect of numerosity may reflect just response biases or the modification of sensory process. This manuscript presented a detailed function of visual and motor adaptation for quantity representation process. The data collected in this study contribute to our understanding how effect adaptation on the numerical processing. However, I

suspect that the sample size seems to be small to evaluate these results. The manuscript could be considered for publication following revisions.

#1

I think it is better to attach the fitting plot from each participant as an appendix. Because of the very small sample size (especially, only 6 participants for visual adaptation condition), it's important to show that the data of adaptation results of each participant to understand the effect size.

We have now increased the sample size, but agree individual results are important, and have now added the fitted plots from each participant in the supplemental material. Thank you.

#2

The major concern is a small sample size. At least, The sample size estimation and power analysis are needed for both conditions (visual adaptation experiment and tapping experiment). I recommend to enlarge sample size.

We agree, and have now done so. We had intended to add even more participants, but Italy was locked down without warning.

#3

Please indicate the number of trials for each participant performed actually for each condition.

Now attended to the manuscript, thank you.

Appendix C

Perth, 29 April 2020

Dear Dr Barton

Thank you and the referees for your work and for accepting this manuscript.

We have dealt with the error in the graph, all should be fine now.

Referee 1 kindly made an optional suggestion for a further test, comparing the effect of adaptation to that of reward. We appreciate this suggestion (which would be very simple to perform, and guaranteed to be significant), but prefer not to include it. We believe that the comparison is not strictly valid, and further statistics could confuse the reader. We prefer to keep the focus on the tests of our specific model, where we find very significant effects.

Thanks again, this is wonderful news during lockdown!

David Burr

For the authors